# Diurnal Variations in Different Precipitation Duration Events over the Yangtze River Delta Urban Agglomeration

**Rui Yao** [1,2,3], **Shuliang Zhang** [1,2,3,*], **Peng Sun** [4,5], **Yaojin Bian** [4], **Qiqi Yang** [1,2,3], **Zongkui Guan** [1,2,3] **and Yaru Zhang** [1,2,3]

1   School of Geography, Nanjing Normal University, Nanjing 210023, China
2   Key Laboratory of VGE of Ministry of Education, Nanjing Normal University, Nanjing 210023, China
3   Jiangsu Center for Collaborative Innovation in Geographical Information Resource Development and Application, Nanjing Normal University, Nanjing 210023, China
4   School of Geography and Tourism, Anhui Normal University, Wuhu 241002, China
5   National Engineering Research Center for Geographic Information System, China University of Geosciences, Wuhan 430074, China
*   Correspondence: zhangshuliang@njnu.edu.cn

**Abstract:** Studying the characteristics of precipitation diurnal variation is beneficial for understanding precipitation formation and underlying mechanisms. In this study, using hourly rain gauge data from 108 stations in the Yangtze River Delta Urban Agglomeration (YRDUA) from 1980–2021, the diurnal variations of the precipitation amount (PA), precipitation frequency (PF), precipitation duration (PD), and precipitation intensity (PI) were analyzed. The effects of elevation, distance of the station from the east coastline, and urbanization on the characteristics of different precipitation duration events were determined. The results indicated that (1) the spatial distributions of PA, PD, and PF were similar in short-duration (SD), long-duration (LD), and ultra-long-duration (ULD), with high values in the south and low values in the north. Most of PA, PD, and PF showed an increasing trend after breakpoint in LD and ULD, but precipitation characteristics in SD showed a decreasing trend before and after breakpoint; (2) the diurnal cycles of PA presented two comparable peaks in the late afternoon and early morning, which occurred SD and ULD precipitation events, respectively. A single peak in the late afternoon (15:00 local solar time [LST]) occurred during the diurnal cycle of PI. The start and peak times occurred mainly in the afternoon for SD and LD. In contrast, the peak time of ULD mainly occurred in the early morning, accounting for 63% of the stations. The start and peak times of LD and ULD occurred in the early morning mainly along the Yangtze River; (3) from the plains to the mountains, the diurnal peaks of PA and PI had gradual variations from noon to afternoon. In addition, dominant diurnal peak values of PA and PI, which are affected by the distance from the east coast, were observed in the early morning in ULD. The effect of urbanization on the difference between urban and rural areas changed from negative to positive after 2000. In addition, urbanization had a significant impact on SD. After 2000, the increase of PA in urban areas was mainly due to the obvious increase of PD and PF in SD, while the increasing trend of LD and ULD in urban areas was smaller than that in rural areas.

**Keywords:** diurnal variation; precipitation duration; urbanization; Yangtze River Delta Urban Agglomeration

## 1. Introduction

The Intergovernmental Panel on Climate Change (IPCC), starting with the Sixth Assessment Report (AR6), is now paying more attention to regional climate change and urban sustainable development issues in conjunction with the Paris Agreement. In line with this, current climate change research should consider the overlapping influences of global, regional, and urban effects on cities. Changes in precipitation distribution and intensity in urban areas are a result of the combined effects of urban heat islands and

the surrounding environment (e.g., ocean and mountains) [1,2]. Diurnal variations in precipitation are the time evolution of atmospheric circulation, with solar radiation as the main driving force [3–5]. The feedback between the daily variations of precipitation and radiation changes the energy budget of the climate system. Studying the characteristics of the daily variations in precipitation is helpful for understanding the mechanisms of precipitation formation and local climate formation. Compared to daily precipitation, one-hour precipitation extremes increase twice as fast with rising temperatures, as expected from the Clausius-Clapeyron relation when daily mean temperatures exceed 12 °C [6]. Short-duration (SD) heavy precipitation is one of the main causes of urban waterlogging. Therefore, understanding the diurnal variations in precipitation and their underlying mechanisms is beneficial for the prediction of regional precipitation and early warning for urban waterlogging.

Previous studies on diurnal variations in precipitation, mainly using surface observations or high-resolution satellite data, showed that summer precipitation or the convective maximum tends to occur in the early morning, late afternoon, or midnight in different climatic zones and landforms [7–11]. For example, early morning rainfall peaks were more common in rainfall events lasting more than 6 h along the western coast of the British Isles. However, rainfall events lasting 1–6 h usually reached the hourly maximum in the late afternoon over the UK [12]. Over the Qilian Mountains in Northwest China, a dominant diurnal peak appears in the late afternoon and an evident second peak in the early morning, respectively [13]. In addition, most stations over the Tibetan plateau exhibit precipitation peaks either in the late afternoon or around midnight [14]. Yu et al. (2007) found that the late-afternoon peak of SD may be explained by the diurnal variation of surface solar heating, which has a great influence on the diurnal variation of low-level atmospheric stability [8]. However, in India, the time of maximum precipitation along the west coast occurs from after midnight to the predawn hours, compared to late afternoon to early evening in the interiors of the subcontinent [15]. Lin et al. (2000) suggested that the nighttime maximum is due to the instability-enhancing stratiform rain due to nighttime radiative cooling at the cloud tops over the world [5]. Moreover, Twardosz (2007a, 2007b) analyzed the role of precipitation type (e.g., frontal and air mass) in forming the general diurnal cycle of precipitation [16,17]. They found that the bimodality of the distribution occurs mainly for air mass precipitation. Chen et al. (2000) suggested that the heavy rainfall of the summer Meiyu Front mostly resulted from well-organized meso-scale convective systems overlapping on the distinctive stratus cloud in China [18].

In addition, precipitation duration is an important factor in measuring temporal and spatial changes in precipitation, and different precipitation durations may be affected by weather systems at different time scales. Precipitation events of different durations present different diurnal features [8,12]. In a study of the warm season over central-eastern China, Yu et al. (2007) found that long-duration rainfall events tended to have their maximum hourly rainfall around early morning, while short-duration rainfall events peaked around late afternoon [8]. It is crucial to determine the effects of precipitation events with different precipitation durations on diurnal variations. The Yangtze River Delta Urban Agglomeration (YRDUA) is one of China's most densely populated and economically developed areas and has experienced frequent and extreme precipitation events and flood disasters [19]. Studies have used daily precipitation data to analyze the trends and spatial distribution characteristics of extreme daily precipitation events over the Yangtze River Delta [20,21]. Previous studies on precipitation over YRDUA mainly concentrated on the daily characteristics of precipitation. However, owing to the highly heterogeneous spatial and temporal distribution of precipitation, daily precipitation data may have discrepancies in the description of precipitation intensity (PI). Moreover, the characteristics of precipitation change with time scales and PI calculated based on daily precipitation data may have deviations [22]. These deviations mainly include overestimating the long-duration precipitation with low PI, underestimating short-duration precipitation with high PI, and dividing the PI with long-duration precipitation events at different times. Therefore, only hourly or higher

time-resolution precipitation data can accurately describe PI and the characteristics of the precipitation evolution process [23,24].

Furthermore, rapid urbanization significantly changes the characteristics of the underlying surface, which has an impact on the diurnal variations in precipitation [25,26]. Urban expansion has progressed rapidly in recent decades in YRDUA. Whether urbanization can affect precipitation has become a research hotspot [27]. During the summer, urbanization has enhanced extreme hourly precipitation in the Yangtze River Delta. During rapid urbanization, Shanghai showed obvious characteristics of a 'Rain Island' and extreme hourly precipitation increased significantly in cities and suburbs. This is mainly the contribution of the urban heat island effect to the extreme hourly precipitation in summer [19,28,29]. Contrary to the interdecadal variability in hourly precipitation and frequency, the intensity of hourly precipitation in the middle and lower reaches of the Yangtze River showed a downward trend from 1991–2004 [30]. Moreover, the diurnal cycle of different precipitation durations in YRDUA, which is commensurate with rapid urbanization processes, has rarely been analyzed. Most previous studies on diurnal precipitation used satellite precipitation products for the calculations, although satellite data uncertainty was not considered [31,32]. Therefore, the average hourly precipitation used in previous studies may not display all aspects of the diurnal cycle [8].

Therefore, this study aimed to analyze the spatio-temporal evolution of the diurnal cycle under three types of precipitation duration events using hourly precipitation data observed at meteorological stations. The effects of urbanization, terrain elevation, and distance from the east coast on the diurnal cycle were investigated. Given the important links of the diurnal precipitation cycle with the prediction of precipitation, early warning for urban waterlogging, and socioeconomic activities, these results may provide significant guidance for urban waterlogging disaster prevention.

## 2. Data and Methodology

### 2.1. Study Area and Dataset

According to the outline of the integrated regional development of the Yangtze River Delta issued by the State Council, the People's Republic of China, the YRDUA consists of 27 cities located in the Shanghai municipality and Anhui, Jiangsu, and Zhejiang provinces (Figure 1). The YRDUA is one of the six largest city groups recognized in the world. It is the largest economic aggregate across China and is experiencing the fastest urbanization. The YRDUA has a subtropical monsoon climate, which is warm and humid with four distinct seasons. The average annual precipitation is 1050 mm, and the average temperature is around 15.5 °C [33]. The precipitation during the flood season (May to July) accounts for 60.78% of the annual precipitation. The seasonal and annual variation of precipitation is very high due to the influence of tropical cyclones in the eastern Pacific and northern Indian Ocean [33,34]. Due to its location in the western Pacific subtropical high pressure, the highest monthly average temperature is greater than 28 °C, and it is one of the warmest regions in mainland China in summer [35,36].

The hourly precipitation dataset was obtained from the National Meteorological Information Center (NMIC) of the China Meteorological Administration. The dataset passed the data quality control by NMIC, including the climate extreme value test, single-station extreme value test, and data consistency test [4]. To avoid the effect of missing data, the missing rate test, the ratio of missing data to total data in a year at the meteorological station, for each meteorological station was calculated during the warm season (from May to September). Years with a data missing rate greater than 2% were defined as a missing year, and rain gauge stations with more than one missing year were excluded [37]. Data from a total of 108 meteorological stations were selected from 1980 to 2021 (Figure 1). China is in the typical monsoon climate zone of East Asia, and strong hourly precipitation mainly occurs during the warm season. However, numerous hazards, such as urban flooding, flash floods, and mudslides, are caused primarily by strong hourly precipitation events. Therefore, this study focused on the diurnal features of rainfall in the warm season. Urban

impervious surface data have previously been used to investigate the relationship between the underlying surface of the city and extreme precipitation indicators, with promising results [38,39]. Therefore, urban impervious surface data at 30-m resolution were obtained from the Tsinghua University website at http://data.ess.tsinghua.edu.cn (accessed on 15 January 2020) [38].

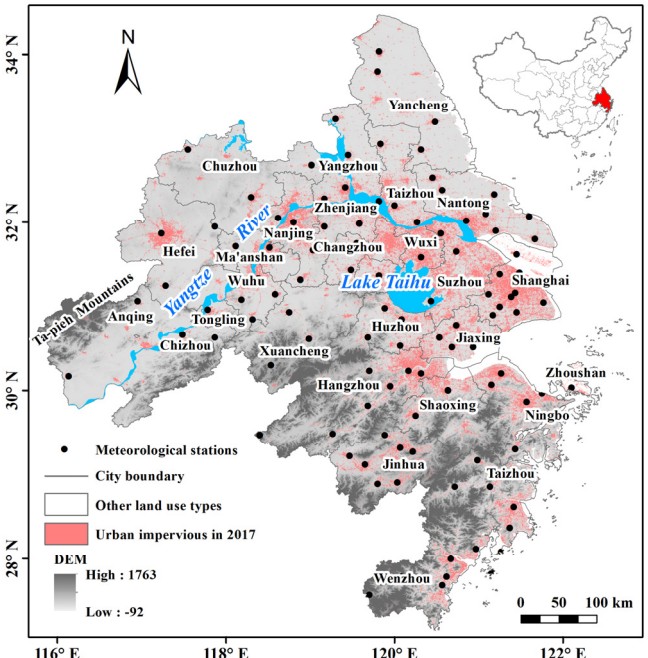

**Figure 1.** Locations of the 108 rain-gauge stations. DEM = digital elevation model.

*2.2. Methods*

2.2.1. The Precipitation Characteristics

The threshold for determining the start of a precipitation amount event is 0.1 mm. Moments when the hourly precipitation is greater than or equal to 0.1 mm are defined as the precipitation time; otherwise, it is defined as no precipitation. The interval between two precipitation events is based on the criterion that no precipitation has occurred for 5 consecutive hours after precipitation time [4]. Therefore, discontinuous non-precipitation times may occur during precipitation events.

The precipitation characteristics in this study included precipitation amount (PA), precipitation frequency (PF), precipitation duration (PD), and precipitation intensity (PI).

(1)   PD: according to previous studies [12], PD is defined as the number of accumulated precipitation hours during a precipitation event. Precipitation events with durations greater than 6 h and those with less than or equal to 6 h are significantly different in terms of diurnal variation [4,40,41]. Based on previous studies [2,13,42,43], this study defined precipitation duration as follows: short-duration (SD), precipitation event lasted for 1–6 h; long-duration (LD), event lasted for 7–12 h; and ultra-long-duration (ULD), event lasted longer than 12 h. Diurnal variations were performed for the three types of precipitation events according to their duration.

(2)   PA: the accumulated precipitation over the total duration of a precipitation event.

(3)   PF: the frequency of occurrence of precipitation events that exceed precipitation of 0.1 mm.

(4)   PI: the total precipitation amount divided by the total duration of all precipitation events recorded at the station.

We used the amplitude (A) of two precipitation features, PA and PI, to describe the diurnal cycle of precipitation. The amplitude was defined using maximum and average values [3,7,16,17]. The formula is:

$$A = \frac{P_{\max} - \overline{P}}{\overline{P}} \cdot 100\% \qquad (1)$$

where $P_{\max}$ is the maximum value of a given precipitation characteristic within 1-h intervals and $\overline{P}$ is the 24-h average value of a given precipitation characteristic: PA and PI. The diurnal cycles of hourly PA and PI in three types of precipitation duration events were calculated in the same way as the total precipitation. For SD, the time of PA and PI only calculated the average of precipitation events less than 6 h.

For each station, we defined the most frequent starting (peak) time of the three types of precipitation events as the time with the maximum number of each type of precipitation event [44]. The proportion of PA in SD (LD, ULD) was the ratio of short-duration (long-duration, ultra-long-duration) precipitation to the total precipitation. In addition, we interpolated the hourly data of 108 meteorological stations using the kriging method to obtain spatial distributions. The correlation was calculated using Pearson's correlation coefficient.

Furthermore, four periods (based on LST) were defined in this study: midnight (from 20:00 to 02:00 LST), early morning (from 02:00 to 08:00 LST), noon (from 08:00 to 14:00 LST), and late afternoon (from 14:00 to 20:00 LST) [43,45,46].

### 2.2.2. Line Trend and Detection of Breakpoint

The Pettitt method is a nonparametric method based on Mann-Whitney [47–49] for the detection of the breakpoint in terms of mean and variance. In this method, it is assumed that two samples, i.e., $x_1, x_2, \ldots \ldots, x_{t0}$, and $x_{t+1}, \ldots, x_T$, are from the same population [50]. if each segment has a common distribution function, i.e., $F_1(x)$, $F_2(x)$, and $F_1(x) \neq F_2(x)$, then the breakpoint is identified at t. In order to realize the identification of breakpoints, the statistical indicators $U_{t,T}$ are defined as follows [47–49]:

$$U_{t,T} = \sum_{i=1}^{t} \sum_{j=t+1}^{T} \text{sgn}(x_j - x_i), 1 \leq t \leq T \qquad (2)$$

where

$$\text{sgn}(\theta) = \begin{cases} 1 & if\ \theta > 0 \\ 0 & if\ \theta = 0 \\ -1 & if\ \theta < 0 \end{cases} \qquad (3)$$

When the time series obeys a continuous distribution, the test statistic $U_{t,T}$ can also be obtained by the following formula:

$$U_{t,T} = U_{t-1,T} + V_{t,T} \qquad (4)$$

For $t = 2, 3, \ldots, T$, where

$$V_{t,T} = \sum_{j=1}^{T} \text{sgn}(x_j - x_i) \qquad (5)$$

The breakpoint $\tau$ needs to meet the following conditions:

$$K_\tau = |U_{\tau,T}| = \max|U_{t,T}| \qquad (6)$$

The $K_\tau$ significance probability is calculated as

$$p = 2\exp\left(\frac{-6K^2}{T^2 + T^3}\right) \qquad (7)$$

Given a certain significance level α, if $p < α$, we reject the null hypothesis and consider $K_\tau$ to be a significant breakpoint in level α.

### 2.2.3. Classification of Stations

Tysa et al. (2019) and Yao et al. (2022) classified meteorological stations in the YRDUA according to the different buffer circulars and proportions of impervious areas [19,51]. The frequency and average precipitation intensity of hourly heavy precipitation in urban areas show distinct urban rain island features [28]. In addition, the strong heat island effect may lead to an increase in the frequency of heavy hourly summer precipitation in the densely populated YRDUA [29]. Choosing optimum reference rain gauge stations is crucial to obtain reliable results for the urbanization effect on hourly precipitation events [52]. In this study, the percentage of impervious area in 5 km buffer circles around the station was used to analyze the urban impact on different precipitation duration events [19,53].

## 3. Results and Discussion

### 3.1. Temporal and Spatial Characteristics in Different Precipitation Duration Events

Different precipitation duration events exhibit different diurnal characteristics and formation mechanisms [8,12]. Figure 2 shows the spatial distribution of the precipitation characteristics for SD, LD, and ULD precipitation events. The spatial of PA, PD, and PF in the three types of precipitation duration events were the same and had low values in the north and high values in the south. In particular, the highest precipitation characteristics in SD, LD, and ULD were observed in the southern coastal areas. Moreover, PF in SD (LD) was the highest (lowest). Compared to SD and LD, the spatial distribution of PI in ULD was slightly different, with high values observed in the northern and western regions. Although PA and PD were four times larger in ULD than in SD and LD, ULD had the lowest PI. In addition, since the PF in SD was three times higher than that in LD, the PA was the same in SD and LD

To further investigate the trends in three types of precipitation duration events, the trends of PA, PD, PF, and PI are presented in Figure 3 before and after the breakpoint. Before the breakpoint, PA showed a slightly increasing trend in the LD (14.5 mm/10 years), but significantly decreasing trends in SD (−16.4 mm/10 years) and ULD (−71.1 mm/10 years). The trends of PD and PF in the three types of precipitation duration events were decreasing before the breakpoint and increasing after the breakpoint. Increasing trends in PI were observed for all three types of precipitation duration events. This demonstrated that the increasing trend of PA in SD cannot offset the decrease in three types of PD caused by the decrease in LD and ULD before breakpoint, which plays a dominant role in the overall PA trend in the warm season.

The proportions of PA, PD, and PF in three types of precipitation events were analyzed to reveal more information about their temporal and spatial differences (Figure 4). The spatial distribution of the proportions in SD and LD were similar, showing high values in the north and low values in the south. On the other hand, ULD showed the opposite spatial distribution, with the proportion being low in the north and high in the south. The results imply that the northern region was dominated by SD and LD, whereas ULD dominated the southern region, with the highest proportion of ULD distributed in the mountains of the southwest. In addition, the proportion of PA and PD in ULD was the highest at greater than 55%. Although PD and PA are essentially the same in SD as in LD, the PF in SD was three times higher than in LD.

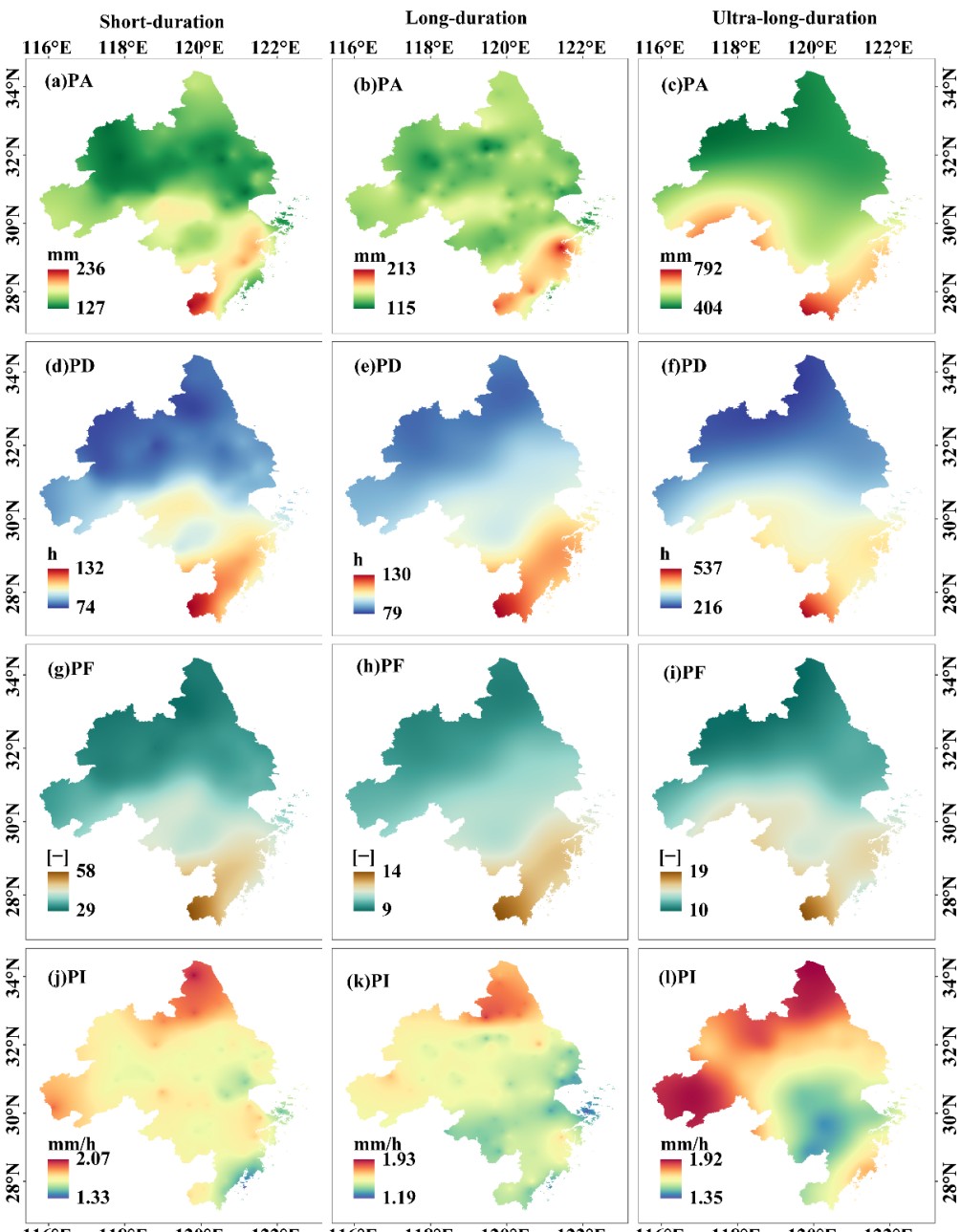

**Figure 2.** Spatial distribution of (**a**–**c**) total precipitation amount (PA), (**d**–**f**) total precipitation duration (PD), (**g**–**i**) total precipitation frequency (PF), and (**j**–**l**) mean precipitation intensity (PI) in short-duration (left panel), long-duration (middle panel) and ultra-long-duration (right panel) precipitation events.

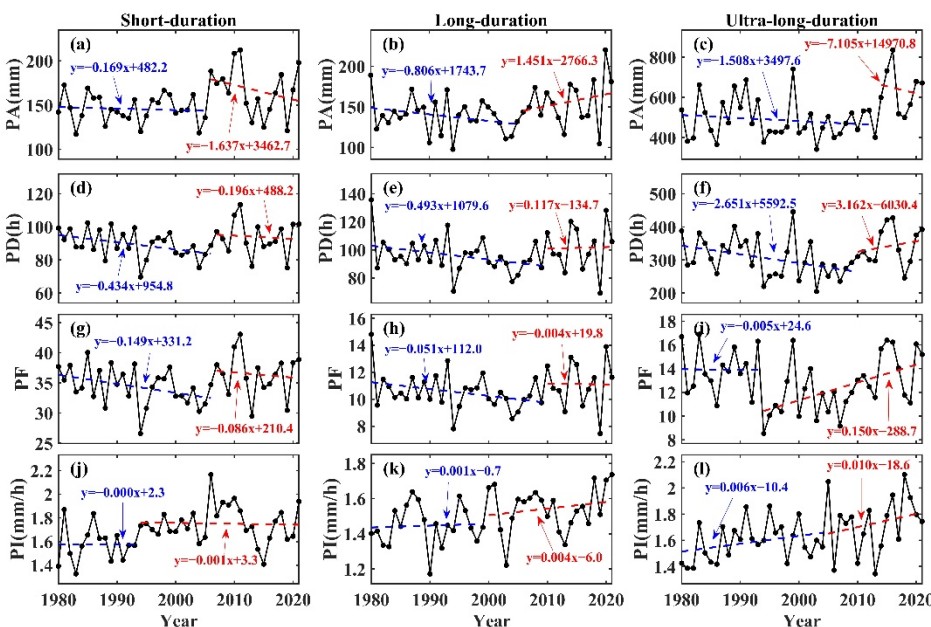

**Figure 3.** Trends of precipitation amount (PA), precipitation duration (PD), precipitation frequency (PF), and precipitation intensity (PI) in short-duration (left panel), long-duration (middle panel), and ultra-long-duration (right panel) precipitation events before (blue line) and after (red line) the breakpoint. (**a**–**c**) are PA in short-duration, long-duration and ultra-long-duration; (**d**–**f**) are PD in short-duration, long-duration and ultra-long-duration; (**g**–**i**) are PF in SD, LD and ULD; (**j**–**l**) are PI in short-duration, long-duration and ultra-long-duration.

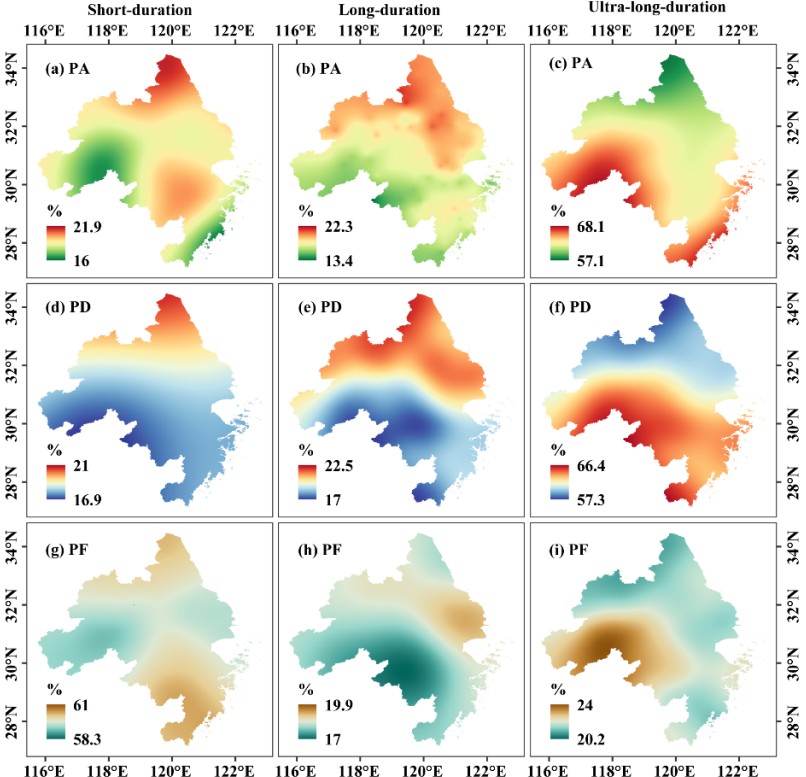

**Figure 4.** Spatial distribution of the proportions of precipitation amount (PA), precipitation duration (PD), and precipitation frequency (PF) in short-duration (left panel), long-duration (middle panel), and ultra-long-duration (right panel) precipitation events. (**a**–**c**) are PA in short-duration, long-duration and ultra-long-duration; (**d**–**f**) are PD in short-duration, long-duration and Ultra long duration; (**g**–**i**) are PF in short-duration, long-duration and ultra-long-duration.

### 3.2. Diurnal Variations in Three Types of Precipitation Duration Events

Understanding the diurnal cycle is important for understanding the precipitation mechanism in the local climate [8,12]. Thus, the amplitude of the diurnal precipitation amount and precipitation intensity is shown in Figure 5. The characteristics of daily PA showed that there were two peaks, one in the late afternoon (16:00 LST) and one in the early morning (06:00 LST) (Figure 5a). The peak value in the late afternoon precipitation was higher than that in the early morning, with precipitation reaching 50 and 43 mm, respectively. PA also showed two minima, one at midnight (24:00 LST) and one at noon (11:00 LST), with the minima at midnight being lower than that at noon. For PI, a single late afternoon peak (15:00 LST) of 1.9 mm/h occurred in its diurnal cycle (Figure 5b).

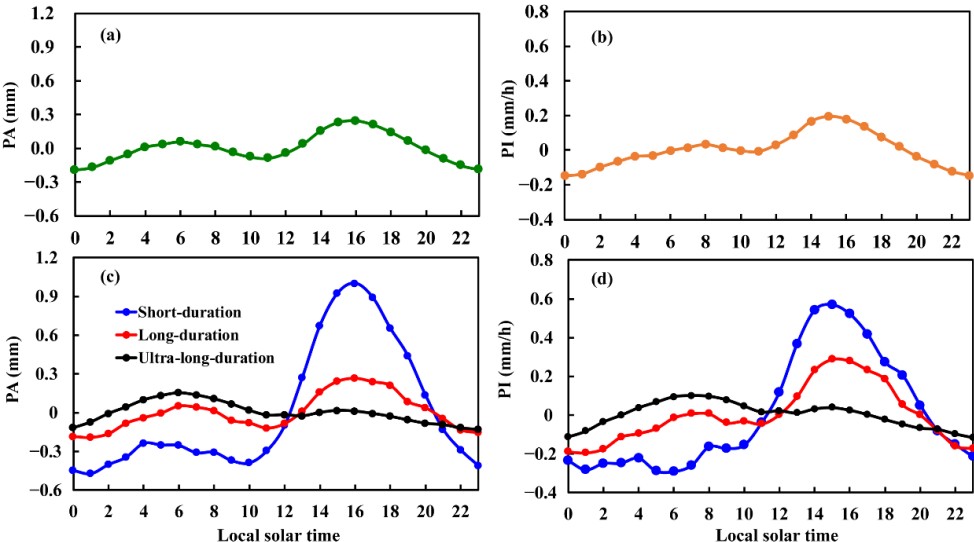

**Figure 5.** Amplitude values of the diurnal cycle in (**a**,**c**) precipitation amount (PA) and (**b**,**d**) precipitation intensity (PI), averaged over YRDUA in short-duration (blue line), long-duration (red line), and ultra-long-duration (black line) precipitation events.

Diurnal precipitation cycles are related to the duration of precipitation events [44]. Figure 5c,d show the diurnal variation in PA and PI for the three types of precipitation events. The maximum diurnal variation in SD and LD occurred in the late afternoon (16:00 LST), with SD having the largest variation at twice the average. Yu et al. (2007) found that the late-afternoon peak of short-duration rainfall events may be explained by the diurnal variation of surface solar heating, which has a great influence on the diurnal variation of low-level atmospheric stability [8]. When the precipitation duration continued to increase (ULD), a second peak in the early morning (06:00 LST) was observed. ULD had maximum hourly precipitation in the early morning, while peak precipitation in SD occurred in the late afternoon. These results further verified the two peaks of average daily precipitation in central and eastern China [8]. Some studies showed that long-duration rainfall is closely linked to the rainfall in the monsoon rain belt, which has been demonstrated to be highly related to the principal frontal zone [34,54,55]. Yu et al. (2007) also revealed that the rainfall in the monsoon rain belt is dominated by long-duration rainfall events with early-morning peaks [8]. The sub-seasonal movement of the monsoon rain belt is mainly contributed by the long-duration rainfall with early morning diurnal peaks. The monsoon rainfall presents alternative early morning and late afternoon diurnal peaks corresponding to the active and break monsoon periods [2,45]. Since low-level clouds often produce small amounts of precipitation, this mechanism could also explain the rainfall frequency being greatest in the morning and intensity being relatively strong in the afternoon. Over the areas covered by the deep continental stratus cloud, the cloud cover hinders solar radiation from reaching the ground and causes relative stability during

the day. However, at night, the cooling of long-wave radiation from the cloud tops causes instability and favors nighttime precipitation [12,42,56].

Figure 6 shows the spatial distribution of PA at different times of the day. PA had the same spatial distribution at midnight and in the afternoon, showing characteristics of low in the north and high in the south, where the high PA values were concentrated in the southern mountainous area. In the early morning and at noon, a diagonal spatial distribution characteristic was observed, and PA was higher in the southwest mountains than in the northeast plains. Looking at PA for the whole season, PA was the highest in the afternoon (average precipitation of 254 mm), followed by early morning (196mm) and midnight (186 mm), and lowest at noon (178 mm). This suggests that PA in the warm season was mainly concentrated in the afternoon (Figure 6).

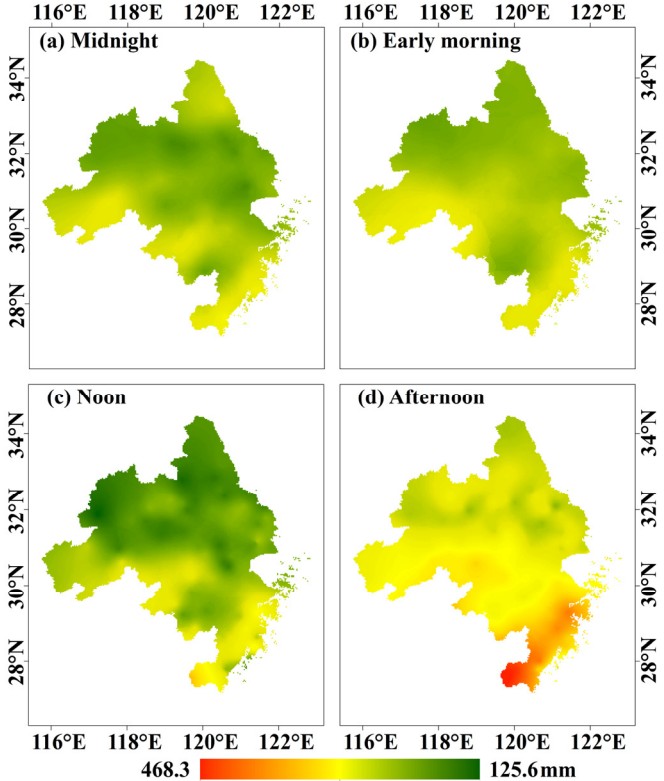

**Figure 6.** Spatial distribution of precipitation amount (PA) in the midnight (from 20:00 to 02:00 LST) (**a**), early morning (from 02:00 to 08:00 LST) (**b**), noon (from 08:00 to 14:00 LST) (**c**), and afternoon (from 14:00 to 20:00 LST) (**d**).

The precipitation peak is closely related to the location and duration of precipitation [12]. For each station, we defined the start (peak) time of the precipitation events as the time when the highest frequency (peak) of precipitation events occurs in diurnal cycles. The start and peak times of precipitation in the three precipitation duration events are shown in Figure 7. The diurnal variation was remarkably different in the three types of precipitation duration events. The start time and peak time occurred mainly in the afternoon for SD, accounting for 86% of stations, respectively (Figure 7a). The start and peak times of SD occurred in the early morning at only 10% of the stations, all located in the northern region. The start and peak times of SD rarely occurred at midnight and noon (Figure 7a,d).

Compared with SD, the start time of LD occurring in the early morning and afternoon accounted for 31%, respectively. The peak time occurred mainly in the afternoon, accounting for 39%, followed by early morning (27%) (Figure 7e). The start time in the southern region occurred mainly in the afternoon or at noon, while it was mainly in the early morning or at midnight in the northern region. Meanwhile, the peak time of LD in the afternoon and early morning had the same distribution as the start time, and the peak time

of LD occurring at midnight and noon had no clustering in space. Similar to SD and LD, the start time of ULD mainly occurred in the afternoon (45%), followed by early morning (26%), and rarely occurred at noon (Figure 7c). In contrast, the peak time of ULD mainly occurred in the early morning, accounting for 63% of the stations, followed by noon at 19% (Figure 7f). The start and peak times of LD and ULD that occurred in the early morning were mainly along the Yangtze River.

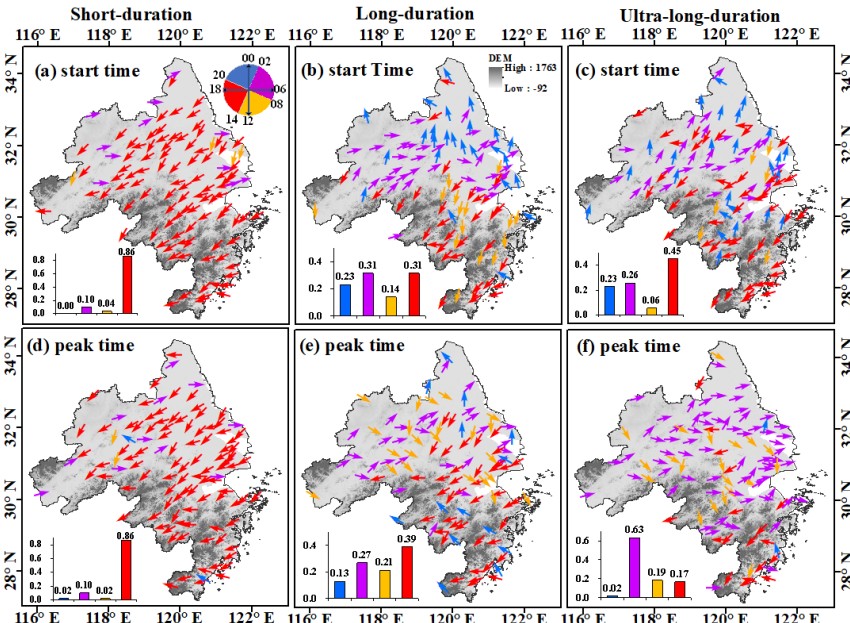

**Figure 7.** Spatial distributions of the diurnal start times (**a–c**) and peak times (**d–f**) in the short-duration (left panel), long-duration (middle panel), and ultra-long-duration (right panel) events. The blue and pink vectors represent the start and peak time between midnight (20:00–02:00) and early morning (02:00–08:00) LST, respectively; the orange and red vectors represent the start and peak time between noon (08:00–14:00) and afternoon (14:00–20:00) LST, respectively.

### 3.3. The Influencing Factors of Diurnal Variation

Studies have shown that elevation, distance from the coast, and urbanization can affect the diurnal variation of precipitation [2,28,32,57]. For example, the average precipitation and frequency in western Sumatra, which is near the Indian Ocean and dominated by highlands, is higher than in the east [32]. Li et al. (2019) found that the precipitation at higher elevations occurred more readily in the afternoon, compared to the lower elevations in the Qilian Mountains [13]. The proportion of the precipitation frequency occurring during the early-morning period decreases with increasing elevations over the two slopes. The urban area has experienced more hourly precipitation extremes than the suburban, and hourly urban precipitation extremes are increasingly inclined to occur during the night [2,28,29]. Therefore, this study attempts to reveal the relationship between terrain elevation, the distance of the meteorological station from the east coastline, and urbanization with hourly precipitation characteristics in three types of precipitation duration events over YRDUA. This should be helpful for a physical understanding of diurnal precipitation variations in this region.

#### 3.3.1. Terrain Elevation

To further statistically analyze the effect of terrain height on PA and PI, we calculated the correlation coefficients between these variables. The correlation coefficients of terrain height with PA and PI were 0.51 and −0.16, respectively, implying an increase (decrease) of PA (PI) with increasing altitude. In addition, the correlation between terrain elevation and precipitation characteristics of SD and ULD exceeded 0.48, which is significantly higher than that of LD. Figure 8 shows diurnal variation and terrain elevation in the

three types of precipitation duration events over YRDUA. Diurnal variations in PA and PI were strongly affected by the terrain elevation. The number of peaks in the precipitation characteristics was not significantly affected by increasing elevation, while the fluctuation and peak value showed a significant increase. For all elevations in SD and LD, the diurnal variations of PA and PI had one dominant peak found in the late afternoon (approximately 16:00 LST) (Figure 8a,b,d,e). For ULD, a dominant diurnal peak appeared in the morning (approximately 08:00 LST). From the plains to the mountains, the diurnal peaks of PA and PI varied from the dominant noon peak to the dominant afternoon peak, and the enhanced afternoon peak was concurrent with the increase in gauge elevation. Moreover, the effect of elevation on the fluctuation and peak value of PA was greater than that of PI. The fluctuation and peak value of the highlands (≥500 m) were larger than those of the lower ground. The peak value of PA in the highlands was almost twice that of the lower ground (Figure 8a–c).

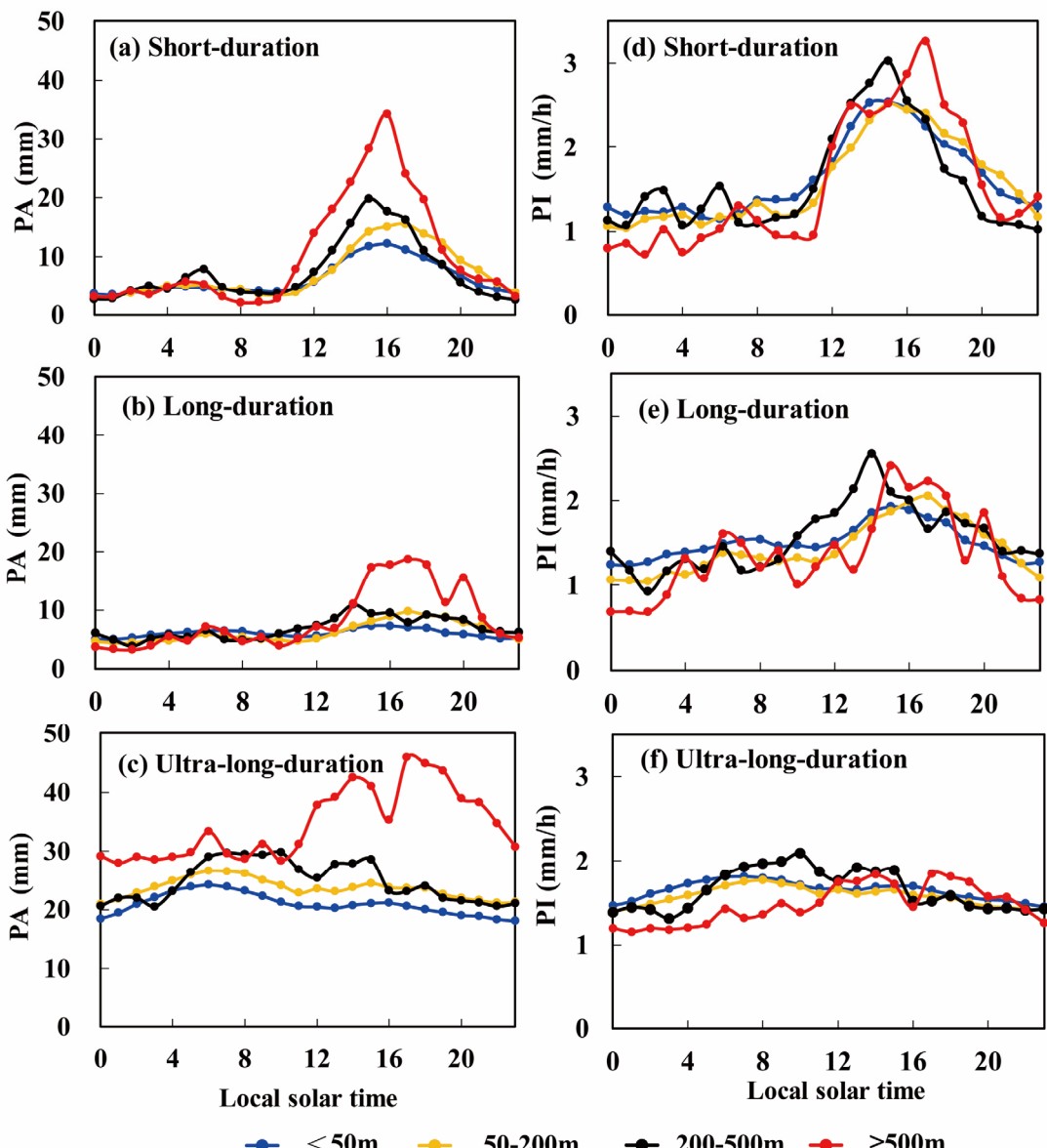

**Figure 8.** Diurnal variation of precipitation amount (PA) and precipitation intensity (PI) with change in elevation for short-duration (top panel), long-duration (middle panel), and ultra-long-duration (bottom panel) precipitation events. (**a–c**) are PA in short-duration, long-duration and ultra-long-duration; (**d–f**) are PI in short-duration, long-duration and ultra-long-duration.

### 3.3.2. Distance from the East Coast

The peak and fluctuation of diurnal variation were affected by the station distance from the east coast, especially for ULD (Figure 9c,f). A dominant diurnal peak of PA and PI was found in the early morning (approximately 08:00 LST) in ULD, which was different from the diurnal peaks of PA and PI affected by terrain elevation. The peaks of PA in SD were not significantly affected by the increase in distance from the east coast, while the fluctuation and peak value of PI showed a significant increase in trend (Figure 9a,d). Some studies have found that the peak of PA occurs in the coastline area and decreases rapidly in stations away from the coast [32]. The most significant fluctuation in PA was observed at station distances of more than 400 km from the east coast and a decreasing PA was observed at stations with decreasing distance to the east coast in YRDUA. The stations far from the east coast had a higher elevation and the uplift of the terrain caused a significant increase in PA and PI.

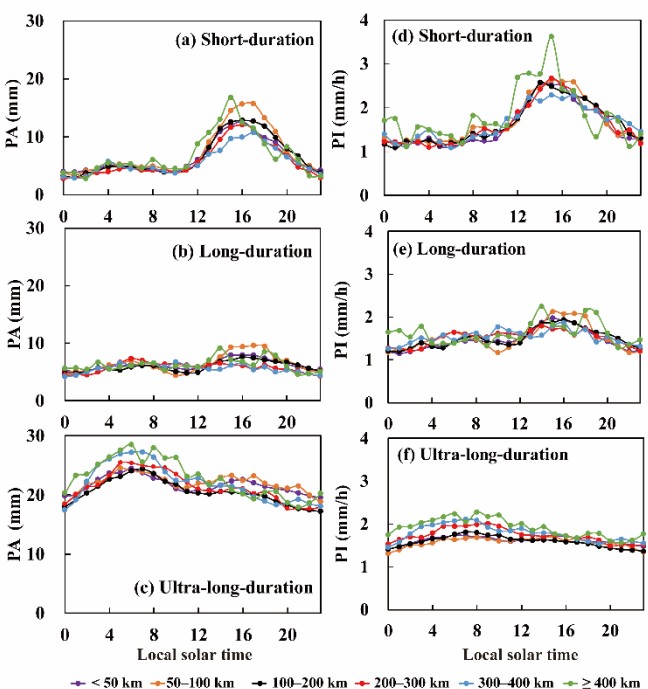

**Figure 9.** Diurnal variation in PA and PI with distance from the east coast for short-duration (top panel), long-duration (middle panel), and ultra-long-duration (bottom panel) precipitation events. (**a**–**c**) are PA in short-duration, long-duration and ultra-long-duration; (**d**–**f**) are PI in short-duration, long-duration and ultra-long-duration.

### 3.3.3. Urbanization

To explore the possible link between precipitation characteristics and urbanization, we compared the precipitation characteristics between the urban and rural stations (Figures 10 and 11). For a given urban area, the precipitation characteristics in the surrounding rural areas were calculated by averaging all the rural station data located in the outer zone extending between 30 and 100 km from the urban area. Rural stations located outside the 100 km threshold were excluded because they might represent distinct climate regimes [19,58]. To avoid the potential influence of elevation on the estimate of the urbanization effect, stations located more than 500 m above sea level were excluded [25,58].

Figure 10 shows the trends of four precipitation characteristics under three types of precipitation duration in urban and rural stations. PA and PI showed increasing trends in urban and rural stations, and the trend of SD in urban was significantly greater than that in rural after 2000. In addition, after 2000, the PI of SD in urban and rural stations showed a decreasing trend, while that of LD and ULD showed an increasing trend. The increasing

trend of PI in urban stations was higher than that in rural stations after 2000. For PD and PF, there was an overall increasing trend. However, before 2000, both urban and rural stations showed a decreasing trend, and the changes were reversed after 2000. From the multi-year average, the PA of urban stations in SD was 14.3 mm higher than that of rural stations after 2000, while the PA of urban stations (0.2 mm) in LD and ULD was lower than that of rural stations (8.0 mm) after 2000. However, the changes in PD and PF were just opposite to those in PI. The PD and PF of ULD in rural stations were higher than those in urban stations, and the PD and PF of SD and LD in rural stations were lower than those in urban stations. Meanwhile, the PI of SD and ULD in urban stations was higher than that of rural stations, which indicates that urban stations have a high frequency of short-duration precipitation and greater precipitation intensity. Zhang and Zhai (2011) found that in southern China, especially in the Yangtze River Valley, the frequency and amount of short-duration extreme precipitation increased during 1961–2000 [59]. Some studies found that three primary effects of urbanization, including urban heat island effects, urban canopy effects, and urban aerosol effects, are potentially altering the PI and PF [2,60–62]. Therefore, after 2000, the increase of PA in urban stations was mainly due to the obvious increase of PD and PF in SD, while the increasing trend of LD and ULD in urban stations after 2000 was smaller than that in rural stations.

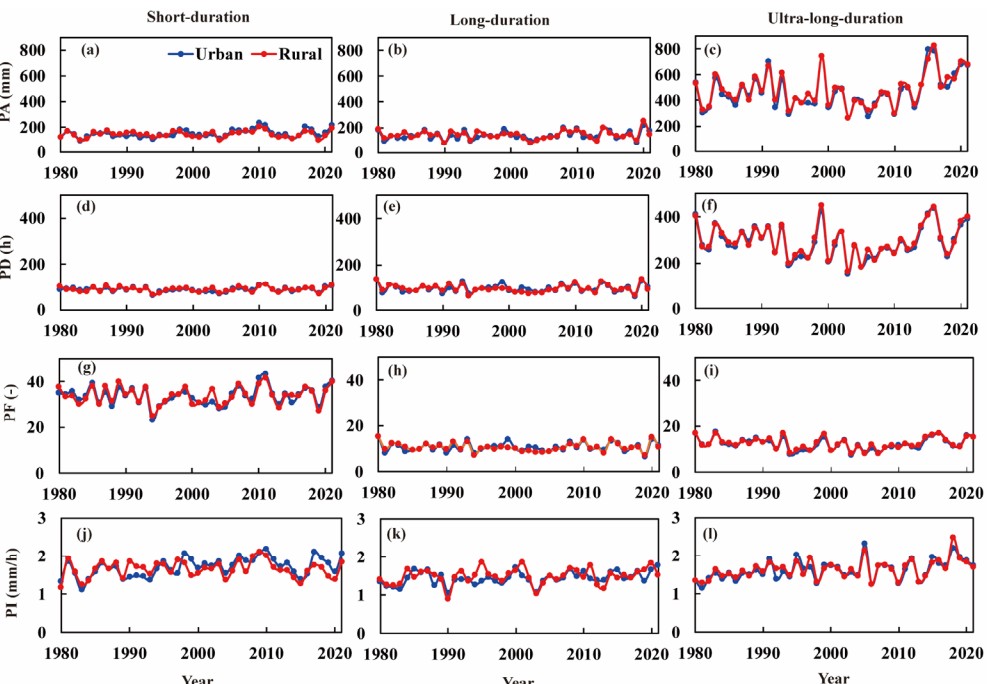

**Figure 10.** PA (**a**–**c**), PD (**d**–**f**), PF (**g**–**i**), and PI (**j**–**l**) during short duration (left panel), long-duration (middle panel), and ultra-long-duration (right panel) precipitation events in urban and rural stations.

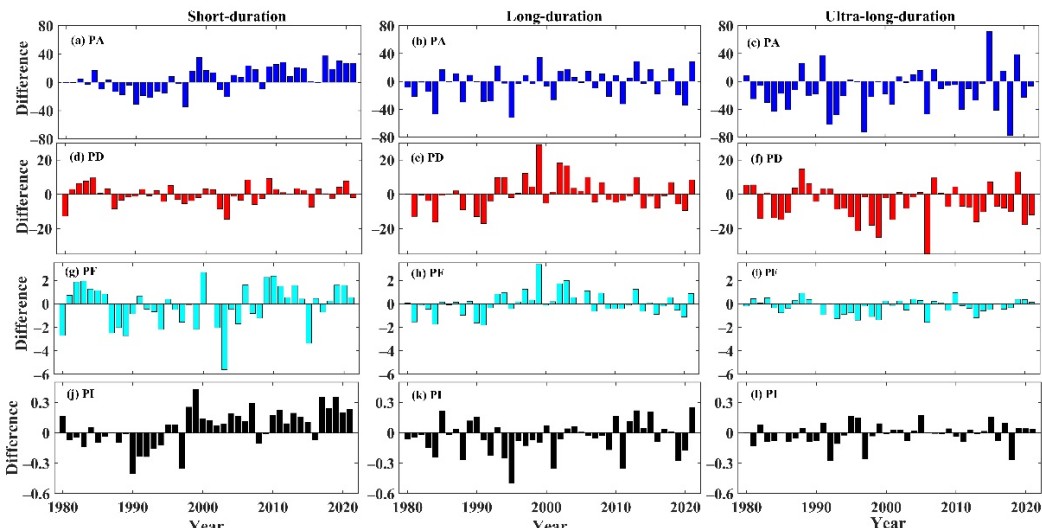

**Figure 11.** Differences between urban and rural stations for PA (**a**–**c**), PD (**d**–**f**), PF (**g**–**i**), and PI (**j**–**l**) during short duration (top panel), long-duration (middle panel), and ultra-long-duration (bottom panel) precipitation events.

The differences in precipitation characteristics between urban and rural stations varied for the three precipitation events. Figure 11 shows that PA and PI in urban stations were lower than those in rural stations before 2000 in SD, implying that heavy precipitation occurred more frequently in urban areas during the rapid urbanization period. With the acceleration of urbanization after 2000, PA and PD were mainly affected by urbanization in SD. Compared to the differences in PA and PI between urban and rural stations before 2000, their average differences increased by 14.9 mm and 0.15 mm/h, respectively, after 2000. This accounts for approximately 9.2% of the average PA and 10.3% of the average PI in their respective rural stations in SD. In contrast, urbanization had less impact on the precipitation characteristics of LD and ULD. The differences in PD and PF between urban and rural stations were opposite to those in PA and PI. PD and PF in urban stations were lower than those in rural stations after the year 2000. Compared to PA in rural stations, PD and PF were higher in urban stations in LD, which led to a higher PA and a lower PI after the year 2000. In addition, the precipitation characteristics in the urban stations were lower than those in the rural stations in ULD.

## 4. Conclusions

Based on the warm season hourly precipitation data from 1980 to 2021, the overall features and regional differences in the diurnal variations of precipitation were investigated over YRDUA. The effects of terrain elevation, the distance from the east coastline, and urbanization were demonstrated. The major results are summarized as follows.

(1) Except for PI, all precipitation characteristics (PA, PD, and PF) had similar spatial distributions in SD, LD, and ULD, with high values in the south and low values in the north. Contrary to the decreasing trend in precipitation characteristics of SD before and after the breakpoint, most of PA, PD, and PF showed an increasing trend after the breakpoint in LD and ULD. SD and LD frequently occurred in the northern regions, while ULD dominated the southern regions, with the highest contribution rate of ULD in the mountains of the southwest.

(2) The diurnal cycles of PA presented two comparable peaks, one in the late afternoon and one in the early morning. A single peak time in the late afternoon (15:00 LST) occurred during the diurnal cycle of PI. The peak time of diurnal precipitation was closely related to the location of the stations and the duration of precipitation events. The peak time of precipitation frequently occurred in the early morning in ULD, while it occurred around late afternoon in SD.

(3) From the plains to the mountains, the peak of PA and PI had gradual variations from noon to afternoon. With increasing elevation, the precipitation peak value increased concurrently in the afternoon. Moreover, the fluctuation and peak value on the highlands (≥500 m) were larger than those on the lower ground. The peak value of PA in the highlands was almost twice that of the lower ground. In addition, the most significant fluctuation was observed at station distances greater than 400 km from the east coast and decreased with decreasing distance to the east coast in YRDUA.

(4) Urbanization affected hourly precipitation characteristics in SD and has less of an impact on LD and ULD. In SD, the effect of urbanization on the difference between urban and rural areas changed from negative to positive after 2000, especially in PA and PI. In contrast, the precipitation characteristics in the urban station were lower than those in the rural station in ULD. After 2000, the increase of PA in urban was mainly due to the obvious increase of PD and PF in SD, while the increasing trend of LD and ULD in urban was smaller than that in rural.

This study analyzed the effects of terrain elevation, distance from the east coast, and urbanization on the diurnal variability of precipitation and obtained interesting conclusions. In the future, we will adopt a different course of action to study rainfall episodes. Based on the analysis of the synoptic situation (e.g., typhoon precipitation and air mass precipitation), it will help to further understand the daily variation of hourly precipitation and provide technical support for the prediction of regional precipitation and early warning for urban waterlogging.

**Author Contributions:** Conceptualization, R.Y., S.Z. and P.S.; Data curation, R.Y., S.Z. and P.S.; Funding acquisition, S.Z., P.S. and Q.Y.; Methodology, R.Y., S.Z., P.S., Y.B. and Q.Y.; Software, Z.G. and Y.Z.; Writing—original draft, R.Y.; Writing—review & editing, S.Z. and P.S. All authors have read and agreed to the published version of the manuscript.

**Funding:** This research was funded by [the National Natural Science Foundation of China] grant number [42271483, 42071364, 42271037, 42101416]; [Key Research and Development Program Project of Anhui province, China] grant number [2022m07020011]; [the Nature Science Foundation for Excellent Young Scholars of Anhui] grant number [2108085Y13]; [The University Synergy Innovation Program of Anhui Province] grant number [GXXT-2021-048]; [Open Research Fund of National Geographic Information System Engineering Technology Research Center] grant number [2021KFJJ06].

**Conflicts of Interest:** The authors declare that they have no known competing financial interests or personal relationships that could have appeared to influence the work reported in this paper.

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
