# Peer review of "Diurnal Variations in Different Precipitation Duration Events over the Yangtze River Delta Urban Agglomeration"

_remotesensing, doi:10.3390/rs14205244_

Round 1
Reviewer 1 Report (Previous Reviewer 3)
I read through the response to reviewers' comments, and the revised manuscript and found that my major comments were not addressed. The authors did not provide answers to my major questions.
I can only repeat my comments from the previous round:
-“Diurnal” means daytime (versus nocturnal, i.e. night-time), while “diel” means a 24-hr cycle (day+night). Please check: https://dictionary.cambridge.org/dictionary/english/diurnal
https://www.merriam-webster.com/dictionary/diel
If the authors mean diurnal, i.e. during the day cycle, then why were midnight and night hours involved in the analysis?
If the authors mean a diel cycle, then how can they analyze events which lasted shorter than 24 hrs?
I think the authors should seriously either rethink or reformulate their study and analysis.
Author Response
Please see the attachment.

Reviewer 2 Report (Previous Reviewer 1)
The authors significantly improved the text. The article could be shortened with weakly related content to the title, as it was written in the previous review.
# 246… total precipitation intensity .. is it not clear?
# 248… extreme is it about heavy?
# 432 The impact of diurnal variation in the three types of precipitation duration events. The title is not clear?
Fig. 7 lacks a description of the X axis.
Round 2
Reviewer 1 Report (Previous Reviewer 3)
This is the revised version of Yao et al. Diurnal variations in different precipitation duration events over the Yangtze River Delta Urban Agglomeration.
I again found that the study did not change, my comments/questions were not answered.
Please find again below a repetition of my suggestions:
-English should be revised. The study contains unclear formulations, incomplete sentences.
-Thank you for providing references from the literature which also use the term "diurnal". Although it is true, that many studies use the word "diurnal" - but unfortunately this does not mean that this is correct. E.g. for Lundquist et al. (2002) Seasonal and spatial patterns in diurnal cycles in streamflow in the western United States - the correct term would be "diel" - as they analysed the 24-hr cycle (diurnal+nocturnal) of streamflow fluctuations in headwater catchments. The best would be if the authors could use correct terminology in their paper. Or provide at the beginning of the paper a definition what they actually mean in their manuscript.
-Introduction: should contain clear description on research gaps in literature and what is new in this study? The goal off the study should be novel. Currently, the aim of the paper is unclear, it is also unclear what is new.
-Methods: unclear section. Based on description, results are not reproducible. Furthermore, description of trend analysis, spatial interpolation, attribution analysis, etc. missing.
-Conclusions are not clear. Abstract lists too many findings - it is not clear what was the aim, and the key findings, and how these are supported by the results.

This manuscript is a resubmission of an earlier submission. The following is a list of the peer review reports and author responses from that submission.
Round 1
Reviewer 1 Report
The title of the article suggests that the authors undertook interesting research that is important not only from a cognitive point of view, but also from a practical point of view. They relate to the 24-hour rainfall cycle of the Yangtze River Delta Urban Agglomeration. After reading the article, however, many serious doubts arise as to the quality of data, research methods and the scope of research. It seems that the authors did not fully rethink their research.
For this reason, I believe that the article should be rejected.
Justification
1. The research covers more than the diurnal cycle. Sections 3.1 and 3.2 are not related to the research topic, and although the purpose is talking about 24-hour cycle fluctuation analysis, it is not in the paper. In these mentioned sections, the authors characterize changes in the amount of rainfall, not changes in the parameters of the diurnal cycle.
2. There is an involuntary statement about data quality, namely # 106-108: "The dataset passed the data quality control by NMIC, including the climate extreme value test, single-station extreme value test, and data consistency test (Yu et al. , 2013). " What amounts of precipitation were the subject of the homogeneity analysis? This is completely unclear. A different procedure for checking the quality of rainfall data applies here.
What was the starting data? Was the data compiled from pluviographic charts? Has the method of measurements changed - e.g. automatic devices have been introduced?
Is the measurement homogeneity of precipitation maintained at 108 stations (there was no change of location and instruments at stations)? Were the sums of precipitation measured from the Hellmann rain gauge compared with the sums obtained from the pluvograph? The phenomenon of lowering the rainfall in records with an analog pluvograph is known, especially at higher rainfall. If there are such differences, corrections should be made to hourly rainfall. Then the rainfall totals are correct. Testing extreme values ​​is probably a misunderstanding, they are measured values!
3. # 131-133: "This study defined precipitation duration as SD if the precipitation event lasted for 1-6 h, LD if the event lasted for 7-12 h, and ULD if the event lasted longer than 12 h)."
How do the authors justify such an artificial division of precipitation events? If it lasts 6 hours, then 1 hour does not rain, and then it continues to rain for 6 hours, according to the authors, it will be 2 rainfall episodes of 6 hours each, or one rainfall of 13 hours. In other words, to study rainfall episodes, you have to adopt a different course of action, first make assumptions about the separation of independent rainfall episodes, e.g. based on the analysis of the synoptic situation.
4. The diurnal cycle has its own amplitude and phase parameters and this should be studied. You can read more about this in the publications, e.g.:
Wallace, J. M. Diurnal Variations in Precipitation and Thunderstorm Frequency over the Conterminous United States. 471 Mon. Weather Rev. 1975, 103, 406-419. https://doi.org/10.1175/1520-0493(1975);
Twardosz R., 2007, Seasonal characteristics of diurnal precipitation variation in Kraków (South Poland), International Journal of Climatology, 27, 957-968.
Twardosz R., 2007, Diurnal variation of precipitation frequency in the warm half of the year according to circulation types in Kraków, South Poland, Theoretical and Applied Climatology, 89, 229-238.
5. # 23 ".. late afternoon peak (16:00 local solar time [LST])"? What does 16:00 mean? Or is it about the hours 4 to 5 pm?
# 112-113 what is the rationale for 1980-2015? Why has the database not been completed until 2020?
# 116-117 .. Error! Hyperlink ..?
# 120 2.2. Methodology. There should be Methods. Methodology is science.
Poorly described methods, the focus was mainly on the description of precipitation characteristics.
# 140-141 Four periods (based on LST) were defined in this study: midnight (from 20:00 to 02:00 140 LST), early Morcing… ”What was the purpose of such an operation?
# 143-144 "Diurnal analysis was performed for the different types of precipitation events according to their duration." How are the types of precipitation defined?
# 162, the research period is missing in caption Fig 2. Moreover, the signature is not clear,
The description of the X axis in Figs. 7 and 10 is different.
Why is the diurnal cycle depicted in Figures from 20:00 LST and not from 00:00 LST. What is the rationale behind this.
Why was the frequency of precipitation only the sum and intensity of precipitation not considered in the precipitation cycle?
There are other ambiguities at work.
Reviewer 2 Report
This study includes the diurnal variations of the precipitation amount (PA), precipitation frequency (PF), precipitation duration (PD), and precipitation intensity (PI) over the Yangtze River Delta Urban Agglomeration (YRDUA) during 1980-2015 summer time. This study also demonstrated the relationship between elevation, the distance of the station from the east coastline, and urbanization with hourly precipitation characteristics in three precipitation duration events (short-duration, long-duration, and ultra-long-duration precipitation events) over YRDUA. Authors need to consider the below-mentioned points before accepting the manuscript to remote sensing journal. The paper could be publishable in Remote Sensing with major revisions.
Major comments:
1. L52-56: “Previous studies on diurnal variations in precipitation, mainly using surface observations or high-resolution satellite data, showed that summer precipitation or the convective maximum tends to occur in the early morning, late afternoon, or midnight in different climatic zones and landforms (Xiao et al., 2018; Hitchens et al., 2013; Muschinski and Katz, 2013; Yu et al., 2007; Luo et al., 2016; Wallace, 1975).”
Please state examples for the early morning (late afternoon/midnight) summer precipitation or the convective maximum and those mechanisms. Readers would like to see more details.
2. L 75-95: The continuity/connectivity seems missing in the introduction section, at lines 75-95. For example, “Precipitation duration is an important factor ….at different time scales” this sentence seems more or less repeated. Please do the necessary modifications to the text in lines 75-95.
3. What is the importance of considering only warm season (April to October) only? Why not other seasons too. The authors should mention the reasons either in introduction or data and methodology sections.
4. What is the purpose and significance of considering the trends for 1980-1999 and 2000-2015 rather than considering complete period 1980-2015?
5. L123-126: The sentence detailing about the precipitation event classification is unclear. Please provide it with clear and concise way.
6. Page 3, section 2.2: In the methodology section, the criteria/conditions for defining precipitation amount (PA), precipitation duration (PD), precipitation intensity (PI), and precipitation frequency (PF) are mostly unclear. The authors should explain all these four parameter in a clear and concise way with appropriate references.
7. L 130-133: The authors should mention the full form for SD, LD and ULD. What is basis/criteria for considering 1-6 h precipitation event as short duration (SD) why not 1 hour precipitation as SD?
8. Figure 2: Please state the mechanisms for high values of PA (PD/PF) over southern regions and low values of PA (PD/PF) over northern regions. Also, why the PI is high over northern regions and low over southern regions? Authors only show the phenomena. Detailed analysis is needed.
9. L228-234; Figure 7: “Diurnal precipitation cycles are related to the duration of precipitation events (Chen et al., 2021). Figs. 7c and 7d show the diurnal variation in PA and PI for the three precipitation events. The maximum diurnal variation in SD and LD occurred in the late afternoon (16:00 LST), with SD having the largest variation at twice the average. When the precipitation duration continued to increase (ULD), a second peak in the early morning (07:00 LST) was observed. ULD had maximum hourly precipitation in the early morning, while peak precipitation in SD occurred in the late afternoon.”
Please state the mechanisms for these phenomena. Also, what is the mechanisms for the development of SD, LD, and ULD (e.g., afternoon thunderstorms, monsoon rainfall, mei-yu rainfall, and etc.)?
10. L282-284: “Studies have shown that elevation, the station distance from the coast, and urbanization can affect the diurnal variation of precipitation (Marzuki et al., 2021; Li et al., 2019; Zhang et al, 2017; Yuan et al., 2020).”
Please give a summary of these previous studies. Readers would like to see more details.
11. Page 13, L 330-332: The authors need to explain in details how they defined the urban area and rural area and their influence.
Minor comments:
1. Authors should provide all the citations as per the remote sensing journal format.
2. L48: “short-duration (SD)” should be “Short-duration (SD)”
3. L148-151: “Except for PI, there was a similar spatial distribution for PA, PD, and PF, with high values in the southern mountainous area and low values in the northern plain area. In particular, the values of PA, PD, and PF were highest in the southern.”
Do you need the 2nd sentence? It is similar to the 1st sentence.
4. Page 4, L 157-158: What is “short precipitation calendar”?
5. Page 6, L 177-178: The sentence at lines 177-178 is more or less duplicate form of the sentence at lines 86-88 “Precipitation events of different durations present different diurnal features and are involved in different forming 87 mechanisms.”
6. Page 7, L 195-195: what is “mm/10a”? Does it mean mm/ 10 years>?! Please mention it clearly.
7. Page 7, L 206-213: The authors need to provide more detailed explanation about “proportion of PA, PD and PF”.
8. Page 9, Figure 7. Is there any reason for starting the x-axis scaling from 20 hrs than 00 hrs?
9. Fig. 8: Please use the same color bar with same value range.
10. Page 9, L 240-248: Please mention the time periods for midnight, early morning, noon and after noon.
11. Figures 10 and 11: Please use same tick mark and value range for y-axis: Figs. 10a,b,c; 10d,e,f; 11a,b,c; 11d,e,f.
12. Figure 12: Please use same tick mark and value range for y-axis: Figs. 12a-c; 12d-f; 12g-I; 12j-l.
13. Figure 12: The urban area increases as the time progresses. How do authors define urban and rural stations from 1980 to 2015?
14. Figure 12: Authors analyze the difference between urban and rural stations for PA, PD, PF, and PI. Please add detailed discussions for pure urban stations and pure rural stations, respectively.
Reviewer 3 Report
The aim of this paper was to analyse the diel variations of precipitation events at the Yangtze River Delta Urban Agglomeration. The authors also determined the influence of elevation, location, and urbanization on the precipitation events characteristics. It was found that the south part of the study area was characterized by higher values of precipitation amount, duration and intensity compared to the south part. The precipitation characteristics showed an increasing trend between 2000-2015. The diel cycles of PA had two peaks in early morning and late afternoon (and these peaks were higher with higher elevation), while PI had only one late afternoon peak. Urbanization affected only short duration precipitation events characteristics.
This is an interesting paper, but there are major methodological and technical issues which I think should be crucial to address.
Specific comments
-“Diurnal” means daytime (versus nocturnal, i.e. night-time), while “diel” means a 24-hr cycle (day+night). I think in this study the correct term would be “diel” instead of “diurnal”. It also does not seem to be correct to analyse diel variations of short and long duration precipitation events if those lasted 6 and 12 hours (according to methods section), respectively – there is no diel, i.e. 24h signal in these cases! I do not understand how the authors could plot such events on Fig 7 to last longer than 6 and 12 hours.
-As far as I understood the authors used only ground precipitation stations – why was this manuscript submitted to the journal Remote Sensing?
-It would be good if the authors could state very clearly what is/are the goal/s of the study, which should be novel – similarly for the abstract and Conclusions section these novel, new findings should be listed. Currently a lot of findings are listed, but many of them are trivial. For instance, higher precipitation amount with higher elevation is not a new finding.
-Study area description is missing.
-Description of methods is vague, the results based on this description are not reproducible. Please see detailed comments in attached annotated pdf, for instance what do the authors mean by “diurnal analysis”; separation of precipitation events is also described in not a clear way, how were the precipitation amount, intensity, etc. values spatially interpolated, trend and breakpoint analysis are not described in the methods, etc.
-Results section: needs major revision. Most methods were not described in the Methods section, it is not clear how these results were obtained, e.g. fig 2 and 4 – how were these values spatially interpolated and averaged? Fig 3 – how was trend and breakpoint analysis performed? Fig 4 – do these differences result from some interpolation artefact? Fig 4 is generally not explained and interpreted in a clear way, this should be revised. Fig 5 presents a lot of very similar figures – I did not understand what was the purpose and message of this figure. Fig 6. – I do not understand what this figure shows, also it is not explained in the methods. Fig 7 – how can the authors plot short duration events for longer than 6 hours if these lasted according to the methods max 6 hours?
-For most figures: legend is either missing or incomplete, it is very hard to understand what each figure shows.
-Instead of plotting many figures, numerical comparisons would be necessary.
-Discussion section better belongs to the results section, while a Discussion section is actually missing.
-The study may benefit from English language editing; there are several unclear formulations, which are very hard to understand.
Technical corrections
-For smaller, technical and more detailed comments please see the attached annotated pdf.
